# High Fecal Prevalence of *mcr*-Positive *Escherichia coli* in Veal Calves at Slaughter in France

**DOI:** 10.3390/antibiotics11081071

**Published:** 2022-08-08

**Authors:** Maryse Michèle Um, Véronique Dupouy, Nathalie Arpaillange, Clémence Bièche-Terrier, Frédéric Auvray, Eric Oswald, Hubert Brugère, Delphine Bibbal

**Affiliations:** 1Institut de Recherche en Santé Digestive, Université de Toulouse, INSERM, INRAE, ENVT, UPS, 31 000 Toulouse, France; 2Innovations Thérapeutiques et Résistances, Université de Toulouse, INRAE, ENVT, 31 000 Toulouse, France; 3Institut de l’Elevage, 75 012 Paris, France; 4CHU de Toulouse, Hôpital Purpan, 31 000 Toulouse, France

**Keywords:** *Escherichia coli*, *mcr* genes, colistin, extended spectrum β-lactamase, critically important antibiotics, healthy veal calves

## Abstract

The aim of this study was to determine the percentage of healthy veal calves carrying *mcr*-positive *E. coli* strains at the time of slaughter in France. Fecal samples were selectively screened for *mcr*-positive *E. coli* isolates using media supplemented with colistin. Screening for *mcr* genes was also carried out in *E. coli* isolates resistant to critically important antimicrobials used in human medicine recovered from the same fecal samples. Overall, 28 (16.5%) out of the 170 veal calves tested carried *mcr*-positive *E. coli*. As some calves carried several non-redundant *mcr*-positive strains, 41 *mcr*-positive *E. coli* were recovered. Thirty-one and seven strains were positive for *mcr-1* and *mcr-3* genes, respectively, while no strain was positive for the *mcr-2* gene. Co-carriage of *mcr-1* and *mcr-3* was identified in three strains. All *mcr*-positive *E. coli* isolates, except one, were multidrug-resistant, with 56.1% being ciprofloxacin-resistant and 31.7% harboring *bla*_CTX-M_ genes. All *mcr-3*-positive *E. coli* carried *bla*_CTX-M_ genes, mainly *bla*_CTX-M-55_. This study highlights the high prevalence of *mcr*-positive *E. coli* strains in feces of veal calves at the time of slaughter. It also points out the multidrug (including ciprofloxacin) resistance of such strains and the co-occurrence of *mcr-3* genes with *bla*_CTX-M-55_ genes.

## 1. Introduction

Colistin is currently considered as the last-resort antibiotic for human infections due to multidrug-resistant *Enterobacteriaceae*, notably those producing extended spectrum β-lactamases (ESBLs) and carbapenemases [1]. Colistin has also been used in veterinary medicine, notably, as a first-intention antibiotic for treating gastrointestinal infections in food-producing animals [2]. The future usefulness of this last-resort antibiotic was challenged by the first description of a plasmid-mediated colistin resistance gene (*mcr-1*) [3], especially as other plasmid-mediated colistin resistance genes (*mcr-2* and *mcr-3*) were also identified [4,5]. Up to the present, seven additional *mcr* genes (*mcr-4* to *mcr-10*) have been identified [6], with *mcr-1* remaining the most prevalent [7]. The most common plasmids carrying the *mcr-1* gene, which are IncI2, IncHI2 and IncX4, have been shown to be highly transferable and to carry other resistance genes [8,9,10]. They contributed to the widespread distribution of the *mcr-1* gene worldwide [6].

As food-producing animals have been identified as a possible source of transmission of colistin-resistant bacteria to humans [6], the monitoring of such strains in the food chain is required. The detection of *mcr*-positive *E. coli* in food-producing animals has been recently reviewed [11]. This review indicated that the comparison of epidemiological studies, and their reported prevalence, is often challenging due to varying study design, methodologies and reporting. The first prevalence data were mainly obtained in a diseased context and through PCR screenings of ESBL-producing *E. coli*, and only a few reports focused on the detection of *mcr* genes in *E. coli* isolated from healthy animals [12]. Moreover, as only a few attempts were made to selectively isolate *mcr*-positive *E. coli*, the true prevalence of such strains in the commensal intestinal microbiota of healthy animals slaughtered for human consumption remains poorly described. This is especially true in the veal calf sector where only a few prevalence studies were conducted [11]. The monitoring of colistin resistance should be carried out in veal calves as they are frequently exposed to antibiotics, including colistin [2,13]. 

The objective of this study was therefore to evaluate the intestinal carriage of *E. coli* isolates positive for *mcr-1*, *mcr-2* and *mcr-3* by veal calves at the time of slaughter in France. A total of 170 fecal samples were selectively screened for *mcr*-positive *E. coli* isolates on colistin-supplemented media. Moreover, as co-detection of *mcr* genes with ESBL genes has been reported [12], ESBL-producing *E. coli* were also isolated from the same 170 fecal samples and screened for *mcr* genes. The occurrence of such genes in *E. coli* resistant to other critically important antimicrobials (CIA) for human medicine: ciprofloxacin and carbapenem was also investigated.

## 2. Results

### 2.1. Detection of mcr-Positive E. coli Isolates

A specific detection of *mcr*-positive *E. coli* isolates in media supplemented with colistin was performed. For 24 out of the 170 calf fecal samples, at least one *E. coli* was isolated from colistin-supplemented RAPID’*E. coli* 2 agar plates which had been inoculated with colistin-supplemented BGBB turbid broths. A total of 27 presumptive colistin-resistant *E. coli* isolates were recovered, as 2 non-redundant isolates were obtained from 3 calves as determined by ERIC-PCR typing (data not shown). PCR screening of the *mcr-1*, *mcr-2* or *mcr-3* gene showed that 23 out of the 27 colistin-resistant *E. coli* isolates were *mcr*-positive (identified as “COL” in Table 1). These originated from 21 calves. Meanwhile, PCR screening of the *mcr-1*, *mcr-2* and *mcr-3* gene was also performed in collections of CIA-resistant *E. coli* isolated from the same 170 fecal calf samples. From these, 110 yielded ciprofloxacin-resistant *E. coli* isolates. ERIC-PCR typing led to the recovery of 172 non-redundant ciprofloxacin-resistant *E. coli* isolates. Twelve *mcr*-positive *E. coli* isolates were identified amongst these strains (identified as “CIP” in Table 1). At least 1 ESBL-producing *E. coli* from 56 fecal samples was isolated. ERIC-PCR typing led to the recovery of 69 non-redundant ESBL-producing *E. coli* isolates, among which 9 were positive for *mcr* genes (identified as “CTX” in Table 1). No *E. coli* isolate was recovered from ChromoID CARBA and OXA-48 media. In all, 44 *mcr*-positive *E. coli* isolates were detected, including 23 isolated from media supplemented with colistin (COL), 12 corresponding to ciprofloxacin-resistant *E. coli* (CIP) and 9 corresponding to ESBL-producing *E. coli* (CTX) (Table 1).

### 2.2. Characterization of mcr-Positive E. coli Isolates

The characterization of *mcr*-positive *E. coli* isolates led to the identification of redundant isolates sharing the same pulsed-field gel electrophoresis (PFGE) type, resistance pattern and *mcr* gene. These were isolated using colistin- and CIP-containing media from three calves (i.e., B2-4, B2-5 and B1-29; Table 1). By contrast, nine calves (i.e., B2-5, A1, A5, B1-6, B1-11, B1-12, B1-13, B1-14 and B1-15) were found each to carry two non-redundant *mcr*-positive *E. coli* isolates recovered from colistin- and CIA-containing media. Lastly, for calf B2-10, two *mcr*-positive *E. coli* isolates belonging to distinct PFGE types were isolated from colistin-containing medium. The three redundant isolates were discarded from our selection of strains, which thus contained 41 *mcr*-positive *E. coli* strains isolated from 28 calves.

Thirty-one and seven *E. coli* isolates were positive for *mcr-1* and *mcr-3* genes, respectively. The remaining three isolates showed a co-occurrence of the two genes (Table 1). All *mcr*-positive *E. coli* isolates, except for one strain (B2-8-COL-1), were multidrug-resistant (MDR), and 56.1% were resistant to ciprofloxacin. None of the *mcr*-positive *E. coli* isolates was positive for plasmid-mediated quinolone resistance (PMQR) genes. ESBL was produced by 13 *mcr*-positive *E. coli* (31.7%), and *bla*_CTX-M_ genes were detected in all these strains. Co-carriage of *bla*_CTX-M-1_ or *bla*_CTX-M-14_ and *mcr-1* was observed in three strains. *bla*_CTX-M-55_ was identified in nine *mcr-3*-positive *E. coli* strains, associated or not with the *mcr-1* gene; *bla*_CTX-M-14_ was identified in one *mcr-3*-positive *E. coli* strain. 

### 2.3. Prevalence of mcr-Positive E. coli Calf Ahedders at Alaughter

Overall, the prevalence of *mcr*-positive *E. coli* in fecal samples from healthy calves was 16.5% (28/170) (Table 2). More precisely, it was 12.9% (22/170) and 3.5% (6/170) for calf carriers of *mcr-1*- and *mcr-3*-positive *E. coli* isolates, respectively. Positive calves came from 10 out of the 32 farms providing calves that were tested at slaughter. Information about the use of colistin could be obtained for 25 farms with only 3 of them reporting the use of colistin. Positive calves originated from two of these farms (Table 2). PFGE analysis showed that *mcr*-positive *E. coli* strains from calves originating from the same farm commonly shared PFGE types, suggesting circulation of strains or clones within a farm and potential transmission between animals. By contrast, PFGE types were distinct between farms, except for PFGE type 1 that was identified in farms 13 and 41 (Table 1).

## 3. Discussion

A significant number of studies on colistin resistance concentrated on pigs and chickens and highlighted high prevalence in these sectors. Fewer studies were conducted in cattle, and they tended to show low prevalence in dairy cattle [11]. The aim of our study was to accurately determine the prevalence of carriage of *mcr*-positive *E. coli* isolates by healthy veal calves at the time of slaughter. To do so, we isolated presumptive colistin-resistant *E. coli* by inoculating BGBB with Durham tubes supplemented with 2 µg/mL of colistin. This isolation procedure was completed by parallel screening for the presence of *mcr* genes in ciprofloxacin-resistant and ESBL-producing *E. coli* recovered from the same fecal samples. Indeed, it has been shown that plasmids carrying *mcr* genes also carried genes encoding resistance to various antibiotics such as cephalosporins, carbapenems and fluoroquinolones [14,15,16]. This multifaceted approach improved the detection of *mcr*-positive *E. coli* isolates from fecal samples. 

We identified that the prevalence of *mcr*-positive *E. coli* in fecal samples from healthy calves was 16.5%, with 12.9% and 3.5% of the calves carrying *mcr-1*-positive and *mcr-3*-positive *E. coli* isolates, respectively. No *mcr-2*-positive isolate was detected here, in agreement with the detection of this gene from animals reported only in China and Belgium [4,17]. In the USA, Meinersmann et al. did not detect any *mcr*-positive *Enterobacteriaceae* when 1077 cecal contents from cattle were tested for presumptive colistin-resistant *E. coli* [18]. A pan-European study from 2008 to 2014 did not report any isolation of *mcr-1*-positive *E. coli* isolates when 3101 cattle were tested [19]. Reviewing the prevalence of *mcr* genes in cattle, Shen et al. reported that the prevalence of *mcr-1* in bacteria was consistently low, from 0.5% to 0.8% in China, 2.6% in Egypt and from 0.1% to 3.3% in Europe [11]. The discrepancies between these studies and ours might partly result from the fact that *mcr*-positive *E. coli* form part of the sub-dominant fecal microbiota. The majority of prevalence studies screened for only a single *E. coli* isolate per animal, consequently, *mcr*-positive strains present in the intestinal microbiota of the animal could remain undetected and are named the ‘phantom resistome’ [20]. In our study, the use of colistin-supplemented media after an enrichment step in the presence of colistin likely increased the identification of *mcr*-positive *E. coli* isolates, which is in agreement with previous reports showing the importance of an enrichment step with colistin to detect *mcr*-positive *E. coli* [21,22]. Screening performed on ESBL-producing or ciprofloxacin-resistant *E. coli* also contributed here to the identification of higher proportions of *mcr*-positive *E. coli* isolates in cattle and also shown by others in France. Haenni and colleagues tested 1398 ESBL producers from diseased veal calves and detected 15.0% and 2.6% of *mcr-1*- and *mcr-3*-positive *E. coli* isolates, respectively [23,24]. These higher proportions are consistent with the fact that these diseased calves might have been treated with antibiotics, and especially colistin. 

Our results confirmed that *mcr*-positive isolates were resistant to other antibiotics [10,25,26]. Except for one isolate, all *mcr*-positive strains were MDR and 56.1% of them were resistant to ciprofloxacin. Regarding resistance to cephalosporins, only 15.8% of *mcr-1*-positive *E. coli* had *bla*_CTX-M_ genes, whereas all *mcr-3*-positive *E. coli* carried such genes. The co-occurrence of *mcr-3* genes and *bla*_CTX-M 55_ was the most frequently observed gene combination. Although the CTX-M 55 subtype is very common in Asia and, until recently, seemed to be limited to this continent [27], the emergence of *bla*_CTX-M 55_ associated with *mcr-1* and/or *mcr-3* genes was reported in France in cattle and in a pediatric infection, suggesting that CTX-M-55-producing *E. coli* could be vectors of *mcr-3* dissemination outside Asia [23,28]. 

Calves harboring *mcr*-positive *E. coli* came from ten farms, from which only two reported the use of colistin in the batch from which the calves sampled at the slaughterhouses originated. Out of the 32 farms, another farm reported the use of colistin, but no calf harboring *mcr*-positive *E. coli* was detected at the time of slaughter. Since the year 2017 when the sampling was performed, the use of colistin in food animal production has drastically decreased in France and several countries around the world, in order to limit the spread of *mcr* genes [29,30]. The occurrence and distribution of *mcr* genes was monitored after the withdrawal of colistin at pig and chicken farms. Studies showed different results, and the association between colistin withdrawal and control of the spread of *mcr* genes was not evidenced [25,31,32,33]. It should be stressed, however, that in the absence of colistin use, treatment with other antibiotics might co-select *mcr* genes and favor their spread. Indeed, eight out the ten farms with *mcr*-positive isolates did not report the use of colistin.

In conclusion, this study revealed a high prevalence of fecal carriage of *mcr*-positive *E. coli* by healthy veal calves at the time of slaughter. It also pointed out the multidrug-resistance of such strains. Although the use of colistin has drastically decreased in food-producing animals in recent years, it should be kept in mind that the use of other antibiotics could lead to the selection of *mcr*-positive isolates. The monitoring of such strains in the food chain is still required.

## 4. Materials and Methods

### 4.1. Study Population and Sampling

Fecal samples were collected from 170 calves aged from 6 to 8 months, originating from 32 calf fattening units and intended for meat production. Feces were collected after evisceration at the time of slaughter in five French slaughterhouses (identified as A, B1, B2, B3 and RA) during 8 sampling campaigns from March to September 2017 (Table 2). A survey was conducted at the farms to identify their use of colistin in the batch from which the calves sampled at the slaughterhouses originated.

### 4.2. Collections of Presumptive Colistin- and CIA-Resistant E. coli

Feces were diluted (10 g in 90 mL) in modified tryptone soya broth (mTSB). In order to detect presumptive colistin-resistant *E. coli*, fecal samples were diluted 1:10 in brilliant green bile broth (BGBB) with Durham tubes and supplemented with 2 µg/mL of colistin. Duplicate BGBB tubes were incubated at 44 °C for 24 h. The positive BGBB tubes were then plated onto RAPID’*E. coli* 2 agar (Bio-Rad^®^) with 2 µg/mL of colistin (COL). At the same time, CIA-resistant *E. coli* were detected by plating fecal suspensions in duplicate onto: (i) RAPID’*E. coli* 2 agar (Bio-Rad^®^) supplemented with 1 µg/mL of cefotaxime (CTX), (ii) RAPID’*E. coli* 2 agar (Bio-Rad^®^) supplemented with 1 µg/mL of ciprofloxacin (CIP), (iii) ChromID CARBA agar and (iv) ChromID OXA-48 agar (bioMérieux^®^, Marcy-l’Étoile, France). All plates were incubated overnight at 44 °C. Up to 8 *E. coli* colonies were collected per positive plate, confirmed for species using the indole production test and typed using the Enterobacterial Repetitive Intergenic Consensus (ERIC) -PCR fingerprinting assay with the ERIC 2 primer [34]. The DNA fingerprints of the isolates collected from each animal were visually compared to identify non-redundant resistant strains in each positive calf that was submitted for further characterization.

### 4.3. Screening for Plasmid-Mediated Resistance Genes

The presence of *mcr-1*, *mcr-2* and *mcr-3* genes was investigated by PCR using previously described primers and conditions [5,35]. For ESBLs producers, multiplex PCR assays were performed to detect (i) *bla*_TEM_/*bla*_SHV_/*bla*_OXA-1_ genes, (ii) *bla*_CTX-M_ genes and (iii) plasmid AmpC genes [36]. The products of the PCR amplification of *bla*_CTX-M_ genes were purified using QIAquick^®^ PCR Purification Kit (QIAGEN) and subjected to bidirectional Sanger sequencing using an Applied Biosystems Analyzer 3130 xl at GeT-Purpan platform (Genotoul, Toulouse, France). The resulting data were analyzed with Sequence Scanner 2 Software and BLAST. Strains resistant to ciprofloxacin were screened for the presence of the PMQR determinants *qnrA*, *qnrB*, *qnrC*, *qnrD*, *qnrS*, *aac(6’)-Ib-cr*, *qepA*, *oqxA* and *oqxB*, with the previously described primers [37,38,39,40,41,42,43].

### 4.4. Screening for Plasmid-Mediated Resistance Genes

Antimicrobial susceptibility was tested by the disk diffusion method and interpreted according to the French Society for Microbiology (SFM) and the European Committee on Antimicrobial Susceptibility Testing (EUCAST) guidelines [44]. The antibiotic disks (Bio-Rad^®^, Marnes-La-Coquette, France) tested included amoxicillin plus clavulanic acid, ampicillin, cefalexin, cefepime, cefotaxime, ceftazidime, cefuroxime, chloramphenicol, ciprofloxacin, ertapenem, gentamicin, nalidixic acid, streptomycin, sulfonamides, tetracycline and trimethoprim. 

### 4.5. PFGE Typing

*E. coli* strains were typed using PFGE on *Xba*I-digested DNA following the CDC-standard operating procedure [45]. Analysis was conducted using BioNumerics software. A Dice similarity coefficient with a UPGMA dendrogram was generated based on 1% tolerance windows and 1% optimization. A cutoff line at 95% was considered to identify genetically related isolates.

## Figures and Tables

**Table 1 antibiotics-11-01071-t001:** Characterization of 44 *mcr*-positive *E. coli* strains isolated from French veal calves at the time of slaughter.

Campaign ID ^1^	Farm ID	Calf ID	Strain ID ^2^	Isolation Medium ^3^	*mcr* Gene	Resistance Pattern ^4^	*bla*_CTX-M_ Gene	PFGE Type ^5^
B2	13	B2-4	B2-4-COL-1 *	COL	*mcr-1*	AMP-STR-TET-CHL-SUL-TMP-CIP-NAL		7
			B2-4-CIP-1 *	CIP	*mcr-1*	AMP-STR-TET-CHL-SUL-TMP-CIP-NAL		7
		B2-5	B2-5-COL-1 *	COL	*mcr-1*	AMP-STR-TET-CHL-SUL-TMP-CIP-NAL		7
			B2-5-CTX-1	CTX	*mcr-1*	AMP-AMC-LEX-CXM-CTX-CAZ-GEN-STR-TET-CHL-SUL-TMP-CIP-NAL	*bla* _CTX-M-14_	16
			B2-5-CIP-1 *	CIP	*mcr-1*	AMP-STR-TET-CHL-SUL-TMP-CIP-NAL		7
		B2-6	B2-6-COL-1	COL	*mcr-1*	AMP-STR-TET-CHL-SUL-TMP-CIP-NAL		7
		B2-7	B2-7-COL-1	COL	*mcr-1*	AMP-STR-TET-CHL-SUL-TMP-CIP-NAL		7
		B2-8	B2-8-COL-1	COL	*mcr-1*	TET		2
		B2-9	B2-9-COL-1	COL	*mcr-1*	AMP-STR-TET-CHL-SUL-TMP-CIP-NAL		7
		B2-10	B2-10-COL-1	COL	*mcr-1*	AMP-STR-TET-SUL-TMP		7
			B2-10-COL-2	COL	*mcr-1*	AMC-AMP-STR-TET-CHL-SUL-TMP-CIP-NAL		1
	14	B2-25	B2-25-CTX-1	CTX	*mcr-1*	AMP-AMC-LEX-CXM-CTX-CAZ-STR-TET-SUL-TMP	*bla* _CTX-M-1_	4
		B2-26	B2-26-CTX-1	CTX	*mcr-1*	AMP-AMC-LEX-CXM-CTX-CAZ-STR-TET-SUL-TMP	*bla* _CTX-M-1_	4
A	32	A12	A12-CTX-1	CTX	*mcr-3*	AMP-AMC-LEX-CXM-CTX-CAZ-GEN-TET-CHL-SUL-TMP-CIP-NAL	*bla* _CTX-M-55_	9
		A15	A15-CTX-1	CTX	*mcr-3*	AMP-AMC-LEX-CXM-CTX-CAZ-GEN-TET-CHL-SUL-TMP-CIP-NAL	*bla* _CTX-M-55_	9
	34	A1	A1-CTX-2	CTX	*mcr-1*, *-3*	AMP-AMC-LEX-CXM-CTX-CAZ-FEP-GEN-STR-CHL-TMP-CIP-NAL	*bla* _CTX-M-55_	11
			A1-CIP-1	CIP	*mcr-3*	AMP-AMC-LEX-CXM-CTX-CAZ-FEP-GEN-TET-CHL-SUL-TMP-CIP-NAL	*bla* _CTX-M-55_	15
		A3	A3-CTX-1	CTX	*mcr-3*	AMP-AMC-LEX-CXM-CTX-CAZ-STR-TET-SUL	*bla* _CTX-M-14_	3
		A4	A4-COL-1	COL	*mcr-1*, *-3*	AMP-AMC-LEX-CXM-CTX-CAZ-FEP-GEN-STR-TET-CHL-SUL-TMP-CIP-NAL	*bla* _CTX-M-55_	15
			A4-CTX-1	CTX	*mcr-3*	AMP-AMC-LEX-CXM-CTX-CAZ-FEP-GEN-TET-CHL-SUL-TMP-CIP-NAL	*bla* _CTX-M-55_	15
			A4-CIP-1	CIP	*mcr-1*, *-3*	AMP-AMC-LEX-CXM-CTX-CAZ-FEP-GEN-STR-CHL-TMP-CIP-NAL	*bla* _CTX-M-55_	15
		A5	A5-COL-1	COL	*mcr-3*	AMP-AMC-LEX-CXM-CTX-CAZ-FEP-GEN-STR-TET-CHL	*bla* _CTX-M-55_	18
			A5-CTX-1	CTX	*mcr-3*	AMP-AMC-LEX-CXM-CTX-CAZ-FEP-GEN-STR-TET-CHL	*bla* _CTX-M-55_	17
B1	37	B1-29	B1-29-COL-1 *	COL	*mcr-1*	AMP-GEN-STR-TET-CHL-SUL-TMP-CIP-NAL		6
			B1-29-CIP-1 *	CIP	*mcr-1*	AMP-GEN-STR-TET-CHL-SUL-TMP-CIP-NAL		6
	39	B1-2	B1-2-CIP-1	CIP	*mcr-1*	AMP-GEN-STR-TET-CHL-SUL-TMP-CIP-NAL		13
	NP2	B1-6	B1-6-COL-1	COL	*mcr-1*	AMP-STR-TET-SUL-TMP		8
			B1-6-CIP-1	CIP	*mcr-1*	AMP-GEN-STR-TET-CHL-SUL-TMP-CIP-NAL		5
		B1-11	B1-11-COL-1	COL	*mcr-1*	AMP-STR-TET-SUL-TMP		10
			B1-11-CIP-1	CIP	*mcr-1*	AMP-GEN-STR-TET-CHL-SUL-TMP-CIP-NAL		5
		B1-12	B1-12-COL-1	COL	*mcr-1*	AMP-STR-TET-SUL-TMP		8
			B1-12-CIP-1	CIP	*mcr-1*	AMP-GEN-STR-TET-CHL-SUL-TMP-CIP-NAL		5
		B1-13	B1-13-COL-1	COL	*mcr-1*	AMP-STR-TET-SUL-TMP		NT
			B1-13-COL-6	COL	*mcr-1*	AMP-GEN-STR-TET-CHL-SUL-TMP-CIP-NAL		NT
			B1-13-CIP-2	CIP	*mcr-1*	AMP-GEN-STR-TET-CHL-SUL-TMP-CIP-NAL		5
		B1-14	B1-14-COL-1	COL	*mcr-1*	AMP-STR-TET-SUL-TMP		8
			B1-14-CIP-1	CIP	*mcr-1*	AMP-GEN-STR-TET-CHL-SUL-TMP-CIP-NAL		5
		B1-15	B1-15-COL-1	COL	*mcr-1*	AMP-STR-TET-SUL-TMP		8
			B1-15-CIP-1	CIP	*mcr-1*	AMP-GEN-CHL-CIP-NAL		5
A2	41	A2-4	A2-4-COL-1	COL	*mcr-1*	AMC-AMP-STR-TET-CHL-SUL-TMP-CIP-NAL		1
	42	A2-22	A2-22-COL-1	COL	*mcr-1*	AMP-STR-TET-SUL		14
		A2-25	A2-25-COL-1	COL	*mcr-1*	AMP-STR-TET-SUL		14
B3-2	50	B3-2-7	B3-2-7-COL-1	COL	*mcr-1*	AMP-STR-TET-SUL-TMP		12
		B3-2-11	B3-2-11-COL-1	COL	*mcr-1*	AMP-STR-TET-SUL-TMP		12

^1^ Sampling campaigns were performed in 2017 as follows: B2, March; B3 and A, April; B1, May; A2, June; B2-2 and B3-2, September; RA, October. ^2^ Redundant *E. coli* isolates originating from the same calf are marked with an asterisk. ^3^
*E. coli* strains subjected to PCR screening for *mcr* genes were isolated on BGBB tubes + 2 µg/mL of colistin and then on RAPID’ *E. coli* 2 agar + 2 µg/mL of colistin (COL), RAPID’*E.coli* 2 agar + 1 µg/mL of cefotaxime (CTX) and RAPID’ *E. coli* 2 agar + 1 µg/mL of ciprofloxacine (CIP). ^4^ AMC, amoxicillin plus clavulanic acid; AMP, ampicillin; CAZ, ceftazidime; CIP, ciprofloxacin; CHL, chloramphenicol; CTX, cefotaxime; CXM, cefuroxime; FEP, cefepime; GEN, gentamicin; LEX, cephalexin; NAL, nalidixic acid; SUL, sulfonamides; STR, streptomycin; TET, tetracycline; TMP, trimethoprim. ^5^ NT, Not Typed.

**Table 2 antibiotics-11-01071-t002:** Distribution of sampled calves according to campaign, farm, number of calves carrying *mcr*-positive *E. coli* and use of colistin in farms.

Campaign ID ^1^	Farm ID	No. of Calves Carrying *mcr*-Positive *E. coli*/No. of Calves Tested	Use of Colistin ^2^
B2	13	7/7	Y
	14	2/3	N
B3	NP	0/5	NK
	29	0/5	N
A	31	0/5	Y
	32	2/5	N
	33	0/5	NK
	34	4/5	N
	35	0/5	N
	36	0/5	N
B1	37	1/6	N
	38	0/4	N
	39	1/5	Y
	40	0/5	N
	NP1	0/4	NK
	NP2	6/6	NK
A2	41	1/5	N
	42	2/5	N
	43	0/5	N
	44	0/5	N
	NP3	0/5	NK
B2-2	45	0/6	N
	47	0/6	N
	48	0/3	N
B3-2	50	2/6	NK
	51	0/6	N
	53	0/3	N
RA	62	0/7	N
	63	0/8	N
	64	0/8	N
	65	0/7	N
	NP4	0/5	NK
Total	32	28/170	

^1^ Sampling campaigns were performed in 2017 as follows: B2, March; B3 and A, April; B1, May; A2, June; B2-2 and B3-2, September; RA, October. ^2^ Use of colistin: Y, Yes; N, No; NK, Not Known.

## Data Availability

Not applicable.

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
