# Peer review of "High Fecal Prevalence of mcr-Positive Escherichia coli in Veal Calves at Slaughter in France"

_antibiotics, 2022, doi:10.3390/antibiotics11081071_

Round 1

Reviewer 1 Report

This is a well-written paper that provides good information about AMR in veal calves. Other than some studies which purposely avoid detecting AMR, these authors used methodology which would increase chances of finding mcr-positive E. coli.  Bravo!  Additional information which would be useful to include, if possible, would be if the veal calves sampled were ever treated with any antimicrobials.  This information would support your hypothesis that use of other antimicrobials might co-select for mcr genes.  Knowing how veal calves are raised, a substantial proportion of them were likely treated with antimicrobials at some time prior to slaughter, but actual data of drug use would be preferable.  A few minor English language issues, but otherwise well done. 

Author Response

We thank the reviewer for this comment. We agree with the reviewer that veal calves sampled in our study might have received antibiotics other than colitsin. Unfortunately, the survey conducted in farms concerned only the use of colistin. We have revised the manuscript to improve English.

Reviewer 2 Report

  • Summary:
    • The authors investigated the carriage of mcr-1, mcr-2, and mcr-3 in veal calves through the isolation and characterisation of E. coli from faecal samples. Through a combination of genotypic and phenotypic susceptibility profiling, PFGE-typing and ERIC-PCR attempted to remove any confounding redundant isolates. The authors have presented their findings clearly and discussed them with relation to other similar studies.

  • Comments

    • Lines 46 – 49: Sentence structure to be revised and made more concise.

E.g. This review indicated that comparison of epidemiological studies, and their reported prevalence’s, is often challenging due to varying study design, methodologies and reporting.

    • Line 54: Consider changing “poorly known” to poorly described or not fully understood.
    • Line 55 and 56, typographical errors:

“This is especially true in THE veal calf….”

“….colistin resistance should be carried OUT in veal calves…

    • Introduction/Objective: Consider introducing which mcr genes and which antibiotics you will be screening for as these do not appear until Table 1 or in Methods (which are at the end).
    • Line 71: E. coli isolates WERE recovered
    • Table 1: Description of campaign ID (as done in Table 2 indicating that this represents the period of sample collection).
    • Results 2.1: Make it clear that 8 isolates were selected for screening from each calf. Revise sentence structure for clarity “at least one isolate from 110 / 170 faecal calf samples”. As 170 faecal samples were used for all parallel screens you could have this as a standalone statement followed by each result.

E.g. Each parallel screening method was performed on 170 calf faecal samples. From these 110 yielded ciprofloxacin-resistant E. coli and 56 ESBL-producing E.coli.

    • Results 2.2: Was redundancy in isolates compared between calves using ERIC-PCR or just for isolates from same calf?

For example, B2-9 has same PFGE type, resistance pattern and mcr-gene carriage as B2-4 and B2-5. As these are the same farm could indicate same strain circulating between animals.

    • Acronyms: Check acronyms (i.e. PFGE, PMQR) throughout as these appear in methods at end and should be specified earlier in text.
    • Result 2.3, Lines 122 – 124: “shared most of the time the same PFGE type” would read better as follows “commonly shared PFGE types”. Consider revising.

End of this sentence (suggesting circulation of strains….) is interpretation and would be better placed in discussion than results.

    • Check spacing between numbers and units. Keep units consistent throughout (e.g. ug/mL or mg/L).
    • Line 142 – 143: Sentence structure – this multifaceted approach improves the detection of mcr-positive E. coli
    • Lines 185 – 186: Change “withdrawn” to withdrawal.
    • Discussion: To support your interpretation that the presence of mcr may be related to co-carriage of other resistance you could review/reference Wang et al. (2020; https://pubmed.ncbi.nlm.nih.gov/32505232/) whom indicate the maintenance of colistin resistance post ban in china due to other antibiotic use.
    • The authors have done a good job of screening for multiple different resistance patterns, these are not discussed in detail in this article as the focus is predominantly on MCR. They may consider Multiple response analysis plots or similar in the future to represent the frequencies of these complex patterns (not required for this report).

Author Response

  • Summary:

The authors investigated the carriage of mcr-1, mcr-2, and mcr-3 in veal calves through the isolation and characterisation of E. coli from faecal samples. Through a combination of genotypic and phenotypic susceptibility profiling, PFGE-typing and ERIC-PCR attempted to remove any confounding redundant isolates. The authors have presented their findings clearly and discussed them with relation to other similar studies.

We thank the reviewer for this comment.

  • Comments
  • Lines 46 – 49: Sentence structure to be revised and made more concise. E.g. This review indicated that comparison of epidemiological studies, and their reported prevalence’s, is often challenging due to varying study design, methodologies and reporting.

Taking into account the reviewer’s comment, we have revised the sentence.

  • Line 54: Consider changing “poorly known” to poorly described or not fully understood.

As suggested by the reviewer, we have replaced “known” by “described.

  • Line 55 and 56, typographical errors: “This is especially true in THE veal calf….”; “….colistin resistance should be carried OUT in veal calves…

Errors have been corrected.

  • Introduction/Objective: Consider introducing which mcr genes and which antibiotics you will be screening for as these do not appear until Table 1 or in Methods (which are at the end).

Taking into account the reviewer’s comment, we have indicated at the end of the introduction which mcr genes and which antibiotics have been screened.

  • Line 71:  coliisolates WERE recovered

The correction has been done.

  • Table 1: Description of campaign ID (as done in Table 2 indicating that this represents the period of sample collection).

The description of campaign ID has been added in Table 1 (footer 1 in the revised Table 1).

  • Results 2.1: Make it clear that 8 isolates were selected for screening from each calf. Revise sentence structure for clarity “at least one isolate from 110 / 170 faecal calf samples”. As 170 faecal samples were used for all parallel screens you could have this as a standalone statement followed by each result. E.g. Each parallel screening method was performed on 170 calf faecal samples. From these 110 yielded ciprofloxacin-resistant  coliand 56 ESBL-producing E.coli.

We took the reviewer’s comment into consideration and accordingly modified this section. Nevertheless, we chose to leave the information concerning the isolation/screening procedure in the materials and methods section.

  • Results 2.2: Was redundancy in isolates compared between calves using ERIC-PCR or just for isolates from same calf? For example, B2-9 has same PFGE type, resistance pattern and mcr-gene carriage as B2-4 and B2-5. As these are the same farm could indicate same strain circulating between animals.

As a first step, for the same calf, ERIC-PCR was used to compare redundancy in E. coli isolated from the same positive plate (RAPID'E.coli 2 agar supplemented with 2 µg/mL of colistin (COL), with 1 µg/mL of cefotaxime (CTX), and with 1 µg/mL of ciprofloxacin (CIP)). It allowed to identify non-redundant isolates for each positive plate that were submitted to further characterization. In a second step, PFGE typing was used to type all mcr-positive E. coli isolates. As mentioned by the reviewer, it highlighted a circulation of strains between calves from the same farm.

  • Acronyms: Check acronyms (i.e. PFGE, PMQR) throughout as these appear in methods at end and should be specified earlier in text.

PFGE acronym appears at the beginning of the 2.2 section and has been specified at this point of the text. PMQR acronym appears in the same section and has been also specified.

  • Result 2.3, Lines 122 – 124: “shared most of the time the same PFGE type” would read better as follows “commonly shared PFGE types”. Consider revising. End of this sentence (suggesting circulation of strains….) is interpretation and would be better placed in discussion than results.

As suggested by the reviewer, the sentence has been revised. Nevertheless, we chose to leave the end of the sentence in this section because we did not discuss circulation of strains in the discussion section.

  • Check spacing between numbers and units. Keep units consistent throughout (e.g. ug/mL or mg/L).

We have checked spacing between numbers and units, and we have replaced “mg/L” by “µg/mL” throughout the manuscript.

  • Line 142 – 143: Sentence structure – this multifaceted approach improves the detection of mcr-positive  coli

The correction has been done.

  • Lines 185 – 186: Change “withdrawn” to withdrawal.

The correction has been done.

  • Discussion: To support your interpretation that the presence of mcrmay be related to co-carriage of other resistance you could review/reference Wang et al. (2020; https://pubmed.ncbi.nlm.nih.gov/32505232/) whom indicate the maintenance of colistin resistance post ban in china due to other antibiotic use.

As suggested by the reviewer, we have added this reference in the discussion (reference 33 in the revised manuscript).

  • The authors have done a good job of screening for multiple different resistance patterns, these are not discussed in detail in this article as the focus is predominantly on MCR. They may consider Multiple response analysis plots or similar in the future to represent the frequencies of these complex patterns (not required for this report).

We thank the reviewer for this comment.

Reviewer 3 Report

The study by Um et al. focuses on the epidemiology of mcr-positive E. coli among healthy veal calves at the time of slaughter in France. This study is very timely as livestock present a serious vehicle of antibiotic spread to humans.

The manuscript highlights carefully the pitfalls of the current epidemiologically studies on the spread of mcr-resistant E. coli in animals, hence making a strong case that this study represents a better designed platform for investigating spread of mcr-carrying E.coli in livestock in France among healthy animals, the latter is an important point as earlier epidemiological studies looked at sick animals.

The approach and the results of this study are described thoroughly. Care was placed not to miss mcr-carrying E. coli strains as a result of possible competition from other gut microbiota they may have a growth advantage over the resistant strains of E. coli; this was done by using colistin in the enrichment step.

Discussion of the results is very comprehensive and well written.

Minor points:

1)      Line 70 is a bit confusing. The use of “was” may need to be replaced by “were”. Also, it is not clear what “as a pair of non-redundant…” refers to.

2)      Line 133, “Less studies…” needs to be replaced by “Fewer studies..”

Author Response

The study by Um et al. focuses on the epidemiology of mcr-positive E. coli among healthy veal calves at the time of slaughter in France. This study is very timely as livestock present a serious vehicle of antibiotic spread to humans.

The manuscript highlights carefully the pitfalls of the current epidemiologically studies on the spread of mcr-resistant E. coli in animals, hence making a strong case that this study represents a better designed platform for investigating spread of mcr-carrying E.coli in livestock in France among healthy animals, the latter is an important point as earlier epidemiological studies looked at sick animals.

The approach and the results of this study are described thoroughly. Care was placed not to miss mcr-carrying E. coli strains as a result of possible competition from other gut microbiota they may have a growth advantage over the resistant strains of E. coli; this was done by using colistin in the enrichment step.

Discussion of the results is very comprehensive and well written.

We thank the reviewer for this comment.

Minor points:

1)      Line 70 is a bit confusing. The use of “was” may need to be replaced by “were”. Also, it is not clear what “as a pair of non-redundant…” refers to.

In this sentence, “was” has been replaced by “were”; and “a pair of” by “two”.

2)      Line 133, “Less studies…” needs to be replaced by “Fewer studies..”

The correction has been done.